# The Intestinal Immune Defense System in Insects

**DOI:** 10.3390/ijms232315132

**Published:** 2022-12-01

**Authors:** Tian Zeng, Saleem Jaffar, Yijuan Xu, Yixiang Qi

**Affiliations:** Department of Entomology, College of Plant Protection, South China Agricultural University, Guangzhou 510642, China

**Keywords:** insect intestinal immunity, physical defense system, Imd pathway, Duox-ROS, JAK/STATpathway, intestinal stem cells

## Abstract

Over a long period of evolution, insects have developed unique intestinal defenses against invasion by foreign microorganisms, including physical defenses and immune responses. The physical defenses of the insect gut consist mainly of the peritrophic matrix (PM) and mucus layer, which are the first barriers to pathogens. Gut microbes also prevent the colonization of pathogens. Importantly, the immune-deficiency (Imd) pathways produce antimicrobial peptides to eliminate pathogens; mechanisms related to reactive oxygen species are another important pathway for insect intestinal immunity. The janus kinase/STAT signaling pathway is involved in intestinal immunity by producing bactericidal substances and regulating tissue repair. Melanization can produce many bactericidal active substances into the intestine; meanwhile, there are multiple responses in the intestine to fight against viral and parasitic infections. Furthermore, intestinal stem cells (ISCs) are also indispensable in intestinal immunity. Only the coordinated combination of the intestinal immune defense system and intestinal tissue renewal can effectively defend against pathogenic microorganisms.

## 1. Introduction

Feeding in insects is an important source of intestinal microorganisms, with large numbers of food-derived microorganisms entering the gut of insects every day, which need to be removed to ensure a healthy gut. Due to the long-term interactions between microorganisms and the gut, the latter has developed a series of defense systems (Figure 1 and Table 1).

The physical structure of the insect gut is the first line of defense against invasion by foreign pathogens [1]. The peritrophic membrane (PM) is one of the important members of the insect intestinal physical defense system, which is mainly composed of chitin and protein [1,37]. The mucus composed of glycosylated proteins is another key physical structure [38]. In addition, the acidic areas of the gut and the intestinal epithelial cells act as natural barriers against pathogens [1,37,38,39]. At the same time, the differentiation in intestinal stem cells provides a continuous impetus for the renewal of the natural barrier [40,41]. The combination of the intestinal physical structure and stem cell differentiation results in a complex and effective physical defense system.

Numerous studies have shown that the immune defense response (Figure 1) is rapidly initiated when intestinal epithelial cells are attacked by pathogens [9,42,43,44,45,46,47], and an imbalance in gut homeostasis can also induce an immune response [48]. The insect gut activates the immune-deficiency (Imd) pathway to produce antimicrobial peptides (AMPs) [49,50,51], which mainly destroy Gram-negative microorganisms. Mechanisms related to reactive oxygen species (ROS) provide a very important line of defense in the midgut immune response and can kill ROS-sensitive microorganisms [52,53]. The janus kinase (JAK)/STAT signaling pathway is also involved in intestinal immune regulation through the production of antimicrobial peptides and the regulation of intestinal stem cell (ISC) differentiation [54] The melanization regulated by prophenoloxidase (proPO) produces a variety of active antimicrobial substances that also have an important role in insect intestinal immunity [55]. Therefore, the intestinal immune defense response is of particular importance in the resistance and elimination of foreign pathogens and in maintaining the stability of the internal environment of the intestinal microbial microbiome.

In this paper, we reviewed the progress of research on insect gut defenses against exogenous pathogens in terms of physical defenses, gut microbes, the gut immune system, and intestinal stem cell differentiation.

## 2. Structure and Function of the Insect Intestine

The insect gut is a tubular structure that connects mouth to anus and runs throughout the entire body cavity [56]. The gut of most adult insects is composed of three structurally, functionally, and developmentally distinct primary domains: the foregut, the midgut, and the hindgut [39,57].

The foregut is located at the most anterior part and includes the pharynx, esophagus, crop, and cardia flap, which have the functions of ingestion, swallowing, chewing, and temporary storage of food [58]. Meanwhile the crop of most insects also has digestive, detoxification and immunological functions, such as in Diptera [45,59]. The cardia flap is the dividing point between the foregut and midgut [55] and can effectively prevent bacteria and particles larger than 0.2 μm from entering the midgut and hindgut by mechanically breaking down food, thereby facilitating host gut defense against insect pathogens [60]. It has also been shown that the cardia in *Drosophila* and *Glossina morsitans* encodes and expresses AMP genes [61,62].

The midgut consists of the peritrophic membrane, the intestinal wall cell layer, the basal membrane, the circular and the longitudinal muscle in order from the inside out [63,64,65,66]. There are four types of intestinal wall cells: enterocytes (ECs) secreting digestive enzymes and absorbing nutrients, enteroendocrine cells (EECs) secreting hormones, intestinal stem cells (ISCs) with differentiation functions, and enteroblasts (EBs) with limited differentiation [66,67,68]. The primary function of the midgut is similar to that of the mammalian stomach, which secretes a large number of digestive enzymes and is an important place for digesting food and absorbing nutrients, as well as being the center of the immune response in the insect gut [64,69,70,71].

The hindgut is connected to Malpighian tubules at the anterior end and consists of the pyloric valve, ileum, colon, and rectum. Its main function is to eliminate the digested food waste from the intestine and recycle water and inorganic salts from the residue [72,73].

## 3. The Physical Defense System of the Insect Intestine

### 3.1. Peritrophic Membrane

The foregut and hindgut of insects are protected by a tightly arranged semi-permeable membrane structure of chitin and protein, known as the peritrophic membrane (PM) [56]. However, the midgut of Hemiptera does not have PM and instead contains a microvillous membrane, the perimicrovillar membrane (PMM) [74]. It is the first physical barrier of the insect intestinal immune system, which prevents damage caused by pathogens, food particles and bacterial toxins ingested by insects and coming into direct contact with intestinal epithelial cells [75,76]. Therefore, the PM is a defense outpost for pathogenic microbial infections that occur in invertebrates via food. As the PM has a role in isolating pathogens, its thickness and integrity are particularly important in defense. Drosocrystallin (dcy) protein is a major component of the PM, and mortality is significantly increased in dcy mutant *Drosophila* following intestinal infection with *Pseudomonas entomophila* [38,77,78]. Moreover, the integrity of the PM may be regulated through the Wnt (Wingless/Integrated) signaling pathway in tsetse flies [79]. It has also been shown that the integrity of the PM is a key factor in the regulation of intestinal homeostasis during changiing in the gut microbiota loads in *Anopheles coluzzii* [80]. In addition, transglutaminase (TG) in *Drosophila* crosslinks with fructose crystal glycoproteins situated on the PM to form stable fibrous structures that more effectively defend against infection by *P. entomophila* in the intestine [77].

### 3.2. Mucus

In many insects, there is a physical barrier between the intestinal epithelium and the intestinal contents, similar to the mucus layer of the vertebrate digestive tract, auxiliary to the PM [2]. Prior to the discovery of the insect gut mucus layer, it was thought that the PM was a structure in the insect gut analogous to the vertebrate mucus layer in protecting cells from acid damage. The mucus is made of mucus-forming proteins (Mf-mucins), which are highly glycosylated proteins found on the surface of epithelial cells lining the respiratory, digestive, and urogenital tracts of vertebrates [39,81]. The physical barrier protects intestinal epithelial cells from infection, dehydration, and physical and chemical damage and facilitates the passage of food through the intestine [81].

More than 30 genes encoding mucin analogs have been identified in *Drosophila*, but little is known about the function of these genes and the role of the encoded mucins [81]. Transcriptome analysis *Drosophila* and *Bactrocera dorsalis* with oral bacterial infection showed that the expression of genes regulating the PM and mucus layer composition was significantly upregulated, suggesting that the insect gut builds two physical barriers in response to pathogenic bacterial invasion [9,82].

### 3.3. Other Physical and Chemical Barriers

The outermost layer of the enterocytes of most insects is also covered with a layer of tightly arranged microvilli, mainly composed of actin [83]. When *Serratia marcescens* and *Bacillus thuringiensis* invade the intestines of *B. dorsalis* and *Leptinotarsa decemlineata*, respectively, the microvilli are abnormally shed as the infection progresses [9,84]. Meanwhile, the genes *Big Bang* and *myosin IB* (*Myo1B*) encode septal junction strength and microvilli structures [44,85].

In addition to the physical barrier of the PM and mucus, there are also physicochemical factors in the intestinal lumen that can degrade pathogens and prevent them from passing through the intestinal epithelium, such as the acidic region of the intestinal lumen [86], digestive enzymes [87], lysozymes [88,89,90], and peptidoglycan hydrolase [91].

## 4. Gut Microbes

The insect gut is an important interface between the host and the external environment, inhabiting a large number of microorganisms. Insect gut microbes play an important role in host nutrition, detoxification, growth, and activation of immune responses [92,93,94]. Meanwhile, in natural open environments, insect feeding is the source of complex and diverse foreign microorganisms including pathogenic microorganisms. Therefore, maintaining the homeostasis of the intestinal microbiome, resisting and eliminating foreign pathogens is crucial for the physiological ecology of insect growth, development, and reproduction.

The gut commensal microbiota of insects is involved in immune defense response, either directly or indirectly. It has been shown that the *Drosophila* intestinal clearance response to viruses relies on gut commensal bacteria to initiate the Imd signaling pathway as a means of activating the ERK signaling pathway in the gut [95]. Recent studies have also shown that the long red cone midge *Rhodnius prolixus* significantly enhances the immune response of the gut upon antibiotic treatment to remove intestinal bacteria by recolonizing *S. marcescens* or *Rhodococcus rhodnii* [96,97]. Ordinarily, intestinal commensal bacteria can protect the host by altering physiological functions, pH, and digestive enzyme levels in the gut and can also eliminate pathogens by competing with them for space and nutrients or by producing antibacterial substances [59]. In mosquitoes, ROS, metabolites, small peptides, and proteins secreted by gut bacteria may directly influence the transmission and development of pathogens [98]. A recent study also confirmed that the *Bombyx mori* intestinal commensal *Enterococcus faecalis* LX10 significantly reduced the germination rate and effectiveness of infection by the microsporidium *Nosema bombycis* [99]. Meanwhile, intestinal commensal bacteria activate the Imd–Relish immune pathway to balance the regeneration of intestinal epithelial cells caused by pathogenic bacteria [100]. The aposymbiotic insects succumbed significantly faster than conventionally reared insects upon bacterial infection in *Rhynchophorus ferrugineus* [101] The endosymbiotic mosquito *Wolbachia* significantly reduces the infection rate of dengue virus and *Plasmodium falciparum*, thus blocking the transmission of dengue fever and malaria [102].

Conversely, some intestinal bacteria secrete substances that inhibit the intestinal immune response and increase the insecticidal activity of pathogenic bacteria. It has been reported that the gut microbe *Citrobacter freundii* of *L. decemlineata* accelerates sepsis caused by *B. thuringiensis* [91]. Moreover, the pathogenic fungus *Beauveria bassiana* interacts with intestinal microbiota to accelerate the death of mosquitoes [103].

## 5. Intestinal Immunity Response

When pathogenic bacteria break through the first line (physical defense system) of the insect gut, the host produces AMPs/ROS-active substances with antimicrobial activity for clearance. Meanwhile, when the gut is attacked by viruses or parasites, it likewise elicits a strong host rejection response, such as activation of the JAK/STAT signaling pathway or the melanization response. In this section, we describe the modulation of multiple pathways activated by pathogens.

### 5.1. Imd Signaling Pathway

AMPs are produced in cellular immunity by two main signaling pathways: Imd and Toll, but it has been shown that AMPs in the insect gut are mainly derived from the Imd signaling pathway [104]. The peptidoglycan recognition protein family (PGRPs), which is subdivided into the transmembrane protein receptor PGRP-LC and the cytoplasmic receptor PGRP-LE, recognizes diaminoylpeptidoglycan (DAP-PGN) or peptidoglycan monomers (TCT) of pathogenic bacteria to activate the Imd signaling pathway in the insect gut (Figure 2) [105,106,107]. PGRP-SD is located upstream of PGRP-LC, and binding to DAP-PGN enhances the recognition signal of PGRP-LC [108]. Recognition of peptidoglycan by PGRPs transmits signals downstream to activate the expression of TAK1 and IKK, and the N-terminus of the NF-κB-like protein Relish is transferred to the nucleus to initiate the expression of AMP genes [3,43,52].

Excess AMPs are detrimental to both the intestine and its normal flora, so the Imd signaling pathway in the gut is regulated by multiple precise and complex negative regulatory mechanisms (Figure 2) [50]. These negative regulators fall into two main categories, one targeting activators of the Imd signaling pathway and one targeting key genes of the Imd signaling pathway. PGRPs with amidase activity, including PGRP-LB, PGRP-SB1, PGRP-SB2, PGRP-SC1a, PGRP-SC1b, PGRP-SC, and PGRPC-SC2, cleave peptidoglycan in the intestine to reduce the source of stimulation of the Imd signaling pathway [10,109,110]. Moreover, Rudra, PIRK, and PGRP-LF and PGRP-LF target the Imd signaling pathway receptors and decrease the number of peptidoglycan recognition receptors [111,112,113]. There are also negative regulatory protein targets that inhibit the Imd signaling pathway, including Dnr1 (Defense repressor 1), which inhibits DREDD activity [114,115]; Caspar, which inhibits the Imd signaling pathway by cleaving relish dependent on DREDD-generated Relish and thus is negatively regulated [116,117,118]; Trabid regulates TAK1 levels [119]; SkpA, which is a subunit of SCF-E3 ubiquitin ligase, targets Relish [120,121]; transcriptional repressors including Caudal [122,123], Oct1 homolog Bbin [124], PP4 (Protein Phosphatase 4) [125] and Myc [126].

### 5.2. Duox–ROS Defense System

In addition to AMPs, ROS also have antibacterial activity in the insect gut [5,127]. Activation of the intestinal nicotinamide adenine dinucleotide phosphate oxidase Duox by pathogenic microorganisms produces ROS that can directly destroy pathogenic bacteria, fungi, or plasmodia [127], thus the DUOX-ROS system plays an important role in insect intestinal immunity. In addition to its involvement in the clearance of pathogenic microorganisms, the Duox-ROS system plays an essential role in maintaining intestinal homeostasis in *B. dorsalis* [128]. Meanwhile, a recent study also showed that serotonin in the gut of *B. dorsalis* and *Aedes aegypti* affects the homeostasis of gut microbiota by regulating the expression of Duox [129].

Many studies have shown that insect intestinal Duox activation produces ROS in two directions, one regulating Duox gene expression in the nucleus and the other activating Duox enzyme activity (Figure 3A). Meanwhile, Duox gene expression is also regulated by two distinct pathways: the MEKK1–MKK3–p38–ATF2 signaling pathway regulates Duox expression in the nucleus [59], and the cell membrane protein Mesh induces changes in Duox expression through an Arrestin-mediated phosphorylation cascade reaction of MAPK JNK/ERK [130]. Furthermore, the MEKK1–MKK3–p38–ATF2 signaling pathway is also dependent on the activation of the PGRPs downstream of peptidoglycan in the Imd signaling pathway, which does not affect the enzymatic activity of Duox [131]. The specific ligand that activates Duox enzymatic activity is uracil, which is secreted by most pathogenic bacteria but not intestinal commensal microbes [132]. It regulates the Hedgehog (Hh) signaling pathway while being recognized by the G-protein coupled receptor (GPCR), activating the formation of Cad99C/PLCβ/PKC endosomes, leading to Ca^2+^ release from the endoplasmic reticulum and activating the enzymatic activity of Duox [133,134].

Peptidoglycan-dependent activation of the Duox transcriptional pathway is negatively regulated by p38 activation, which itself is regulated by PLCβ, Calcineurin B (CanB) and MAP kinase phosphatase 3 (MKP3) (Figure 3A) [135]. This negative regulation ensures that transcriptional Duox is activated only when stimulated by large amounts of peptidoglycan, thus being able to protect the normal proliferation of the intestinal flora. ROS scavenging mechanisms exist in the insect gut, as excess ROS cause oxidative stress damage to intestinal epithelial cells. ROS enzymes are mainly involved in regulating normal levels of ROS that maintain normal oxidative reactions in the intestine and avoid cell damage by the excess of ROS such as catalase [127], long-oxide dehydrogenase, thioredoxin peroxidase, and glutathione peroxidase [12]. The Nrf2/Keap1 signaling pathway activates the production of the abovementioned ROS enzymes when intestinal epithelial cells are exposed to oxidative stress [136,137]. Upon an invasion of exogenous pathogens in the gut, Duox regulates ROS in order to clear the overexpressed ROS in the immune response by various peroxidases in vivo to maintain their levels within the threshold of host damage.

### 5.3. JAK/STAT Signaling Pathway

In conjugation with the Duox and Imd signaling pathways which are two major complementary immune defense pathways in the insect gut, there are other pathways that activate AMPs to participate in the immune defense response, such as the activation of the midgut JAK/STAT pathway that induces *Drosomycin* expression to defend against fungal invasion [138,139,140,141]. The pathway is conserved throughout biological evolution and plays an important role in insect innate immunity [141,142]. Numerous studies have revealed its involvement in intestinal immune responses, such as resistance to viral infections [143,144,145,146] and resistance to fungal and bacterial infections [147,148].

Regulation of the JAK/STAT signaling pathway is mediated by a variety of cytokines that regulate many important biological processes such as immune regulation, cell proliferation, differentiation, and apoptosis [149]. The JAK/STAT signaling pathway is activated by the binding of secreted ligands to receptors, leading to the aggregation of JAK and the activation of STAT by phosphorylation and subsequent translocation to the nucleus to regulate the expression of target genes. In contrast to the multiple JAK/STAT combination pathways in mammals, only one typical pathway has been identified in *Drosophila* [150,151]. It has three main components: the receptor Domeless [152,153]; the JAK Hopscotch [154] and the transcription factor STAT92E [155,156]; Domeless is regulated by an unpaired (Upd) family of three secreted proteins, Upd1, Upd2, and Upd3 [157]. The pathway is also regulated by a number of negative feedback regulators, including Socs36E, Ptp61F, and Wdp [158,159,160].

Numerous studies have revealed an important role of the JAK/STAT pathway in the insect intestinal immune response (Figure 3). Starting with the observation of phosphorylation and translocation of STAT92E in *Anopheles gambiae* infected with enteric bacteria [161], there has been growing evidence of its involvement in the insect intestinal immune response. Intestinal infection with the pathogens results in damage to the intestinal epithelium, leading to the release of the ligands Upd2 and Upd3 that activate the expression of *Drosomycin* (Figure 3B) [138,162]. Contact between common microbes and intestinal commensal microorganisms also activates the JAK/STAT pathway to enhance immunity in the gut [163]. Furthermore, it has also been shown that the production of Upd3 and activation of the JAK/STAT pathway in the gut must also depend on the oxidative burst of Duox (Figure 3A) [164]. Meanwhile, the JAK/STAT pathway has a crucial role in host defense against viral infection according to activating the expression of IFN-stimulated genes (ISGs), which are a group of secreted proteins that play key roles in innate immunity [165,166].

### 5.4. Melanization in the Insect Gut

Insect intestinal immunity also includes a melanization response activated by the propheoloxidase activating system (proPO system) (Figure 1) [64,167]. It is triggered by the activation of proPO enzymes through the protein hydrolysis of a series of clip domain serine proteases, which produce polyphenol oxidase (PPO) that oxidize phenolic compounds such as tyrosine and dopa to quinone, which is further converted to melanin [168,169]. The activation process of melanization produces many antimicrobial active substances with bactericidal properties that enable the clearance of pathogenic bacteria in intestinal infections [170].

The melanization response contributes to wound healing by isolating dying pathogens from surrounding tissues and is therefore primarily associated with death or tissue damage [40,171,172]. Current research on the origin of PPO has led to the following findings: the midgut PPO of adult mosquitoes probably originates from the hemolymph [171,172,173]; the presence of PPO is not detected in the midgut of the silkworm, but is found in the hindgut, which is therefore the center of melanization [174]. Pathogenic microorganisms enter the hindgut from the midgut and before being eliminated from the body, there is a PPO bactericidal response in the intestine, which becomes the last line of immune defense of the insect gut.

However, *B. thuringiensis* can inhibit the melanization reaction by producing Bt protein crystal protein 1Ab (Cry1Ab), leading to an increase in intestinal flora density in the hindgut of silkworms [175]. Microbes in the midgut of *Helicoverpa armigera* also mediate the downregulation of the antiviral factor PPO to promote baculovirus infection [176].

### 5.5. Intestinal Immunity against Parasitic and Viral Infections

Inclusive of bacterial infections in the insect gut, there are also microsporidia and viruses that parasitize the intestinal epithelium and cause infections [177]. As with pathogenic bacteria, the entry of a virus or parasite into the insect’s gut exposes it to the physical defenses of the gut, the first line of defense, as well as activating the immune response of the intestinal epithelium.

Infection of *B. mori* larvae with the silkworm microsporidian *N. bombycis* activates the midgut Toll pathway and the JAK-STAT pathway to produce antimicrobial peptides, and melanization in the hemolymph of silkworm larvae was also found to be inhibited [178]. This suggests that there is information transfer between the hemolymph and the gut when the insect gut is infected. When the intestinal parasite melanizes in mosquitoes, it is retained throughout the life cycle of the insect [172]. Moreover, after the intestinal tract of the bumblebee *Bombus inptiens* is infected with the parasite *Crithidia bombi*, the central nervous system pathway will be destroyed, and cognition and behavior change, indicating that there is brain–gut axis communication [179].

Viruses enter intestinal epithelial cells through virus receptors in intestinal microvilli, causing a host immune response [180]. Bmlipase-1 (Bm serine protease-1) [181], BmSP-2 (Bm serine protease-2) [70], and alkaline trypsin in the gut of the silkworm have strong antiviral activity against the nucleopolyhedrovirus (MNPV) [182]. Analysis of the midgut transcriptome of silkworms infected with *Bombyx mori* cytoplasmic polyhedrosis virus (BmCPV) revealed increased expression of the serine protease inhibitor Serpin-5 [183]. Additionally, infection of *Autographa californica* with AcMNPV regulated expression of p35 and inhibited apoptosis, suggesting that host cell apoptosis is also a strategy to combat virus infection [184]. Meanwhile, studies of insect intestinal resistance to viral infestation have shown that the Imd signaling pathway is also involved in the antiviral response [96]. When viruses enter the *Drosophila* intestine, the PGRP-LC receptor on the intestinal epithelial cell membrane recognizes peptidoglycans of the microorganis and activates the Imd signaling pathway, which initiates NF-κB signaling and activates the expression of Pvf2 ligands, thereby activating ERK antiviral activity in the intestine through the binding of Pvf2 ligands to the PVR receptor [95].

## 6. Gut Renewal

The intestinal epithelium of adult insect, such as vertebrates, is a highly regenerative tissue that rapidly self-renews in response to changing inputs from nutrition, gut microbiota, ingested toxins, and signals from other organs [185]. The regeneration of insect intestinal epithelial cells is controlled by the proliferation and differentiation of ISCs, which play a replenishing role in intestinal defense against pathogen invasion (Figure 3B).

It has been reported that intestinal epithelial cells of adult *Drosophila* are controlled by the proliferation and differentiation of ISCs, thereby controlling the rate of regenerative renewal [65]. When pathogenic bacteria or toxic substances invade the insect intestine and cause acute damage to intestinal cells, *Drosophila* regulates intestinal epithelial cell structure and function by altering the rate of ISC proliferation and differentiation to repair intestinal damage [186]. Signaling pathways such as JAK/STAT, EGFR, Hippo, and JNK have been reported to be involved in regulating ISC differentiation [162,187,188,189].

Interestingly, recent studies suggested that secondary metabolites of enteric microbiota also activate the differentiation of ISCs [190]. Meanwhile, gut microbes promote ISC differentiation into enterocytes (ECs) via the microbial pattern recognition pathway Imd/Relish, whereas pathogenic bacteria promote ISC differentiation into enteroendocrine cells (EE) via the JAK/STAT signaling pathway [100]. It has been found that an increase in the number of gut bacteria in Imd-mutant *Drosophila* caused an accelerated rate of ISC division, whereas ISC proliferation and differentiation was inhibited in Duox-mutant *Drosophila*, suggesting that the ROS molecules produced by Duox are key factors in the induction of ISC division [82]. Excess ROS in the intestinal immune response damages intestinal epithelial cells, which induces ISC proliferation and differentiation (Figure 3A).

Therefore, intestinal epithelial cell differentiation is very importantly linked to the intestinal immune system, and only the immune response together with intestinal stem cell differentiation can work together to fight pathogenic bacteria quickly and efficiently [191].

## 7. Conclusions

Exogenous pathogens that enter the insect gut are intercepted and eliminated by a variety of factors. Upon pathogen invasion, intestinal epithelial cells are protected by different means such as the PM, mucus layer, acidic zone, and various digestive enzymes. These cells can recognize pathogenic bacteria and activate the production of bactericidal effector molecules such as AMPs and ROS in the intestine, thus eliminating harmful microorganisms. Nevertheless, Imd and JAK/STAT signaling pathways produce the AMPs, Duox/Mesh, which mediate the expression of ROS and melanization cascades involving numerous diverse bactericidal substances that also play important roles in intestinal immunity. Enhancing the ability of intestinal epithelial cells to proliferate and differentiate is also a way to defend against foreign pathogens. The entire intestinal immune defense system helps the intestine to fight off pathogens very effectively.

An immune homeostatic mechanism exists in the insect gut that ensures effective defense against foreign pathogen invasion while protecting the gut from damage caused by immune overload. This homeostasis, developed during the long evolution of insects, has been a hot topic of research in recent years. Many studies have demonstrated multiple mechanisms in host insects to ensure a moderate immune response, ensuring an appropriate immune response to pathogenic bacteria and the regulation of gut flora homeostasis while avoiding an overexcited immune response.

The study of insect intestinal immune responses has provided a theoretical basis for opening up new technologies for agricultural pest control. In recent years, many studies have focused on the killing power of bacterial functional molecules, while less research has been carried out on the disordered killing power of insect intestinal immune homeostatic mechanisms. The lethal role of disordered intestinal immune homeostasis in pest control has broad application prospects, and the use of genetic engineering to modify insects while exploring new homeostatic mechanisms will be one of the research hotspots in coming years.

Although much has been achieved in the study of the insect intestinal immune system, many open questions remain. The Imd signaling pathway and Duox-ROS defense system have been shown to activate ISC differentiation and enhance the renewal rate of intestinal epithelial cells in addition to removing pathogens. This demonstrates that the regenerative renewal capacity of intestinal epithelial cells is particularly important for insects to defend themselves against pathogenic microorganisms. At the same time, there may be additional signaling pathways that jointly regulate ISC differentiation. It further remains unknown or what measures insects use to coordinate the direction of ISC differentiation and competition for differentiation. We believe that this will be the focus of future research. In addition, research on the brain–gut axis (the regulatory system for two-way communication between the nervous and gastrointestinal systems) has received attention in recent years. Communications between the intestinal immune system and the nervous system in insects lead to changes in activity and behavior, which may also provide more possibilities for the future application of intestinal probiotics in human medicine.

## Figures and Tables

**Figure 1 ijms-23-15132-f001:**
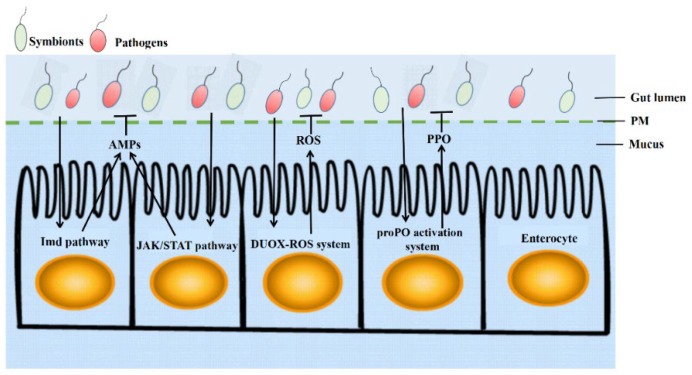
The insect intestinal immune defense system. The insect intestinal immune defense system consists of a physical barrier and an immune response. Pathogens entering the intestinal epithelium face a physical barrier such as the mucus layer of the peritrophic membrane and intestinal cells. A number of signaling molecules produced by pathogens activate a range of immune responses: the Imd signaling pathway and the JAK/STAT signaling pathway produce AMPs, Duox produces ROS, and the melanization response produces a range of bactericidal active substances. This whole system effectively regulates the balance of gut microbes. PM: peritrophic membrane; AMPs, Antimicrobial peptides; ROS, reactive oxygen species; PPO, polyphenol oxidase.

**Figure 2 ijms-23-15132-f002:**
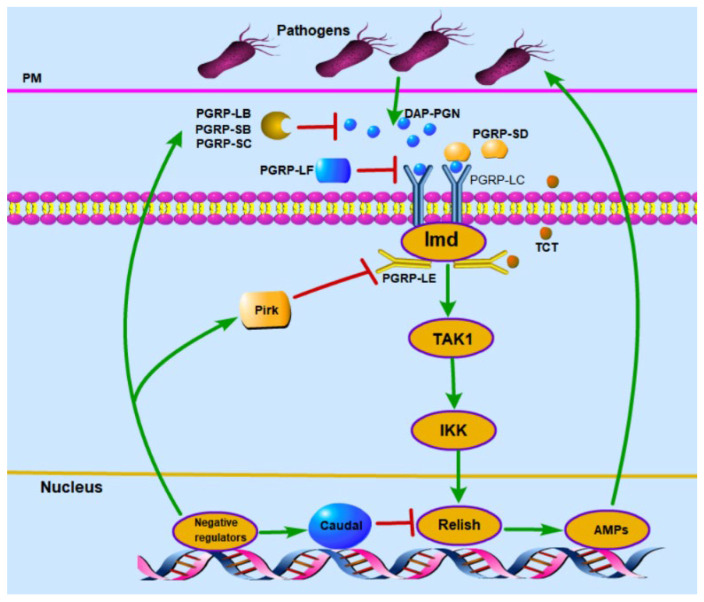
Imd signaling pathway and its negative regulator in insect intestinal immunity (modified from the reference [59]). When pathogens invade the gut, diaminopyrimidine-type peptidoglycan (DAP-PGN) in the cell wall is recognized by the peptidoglycan recognition receptor LC (PGRP-LC) in the cell membrane, activating the intranuclear transcription factor Relish to regulate antimicrobial peptides (AMPs); the intramembrane receptor PGRP-LE recognizes tracheal cytotoxin (TCT) and activates Imd pathway; Pirk is a repressor of PGRP-LE and PGRP-LC activity, PGRP-LF blocks PGRP-LC activity, and Relish transcription is regulated by negative feedback from Caudal genes. tAk1, transforming growth factor-β-activated kinase 1; IKK, inhibitor of kappa B kinase.

**Figure 3 ijms-23-15132-f003:**
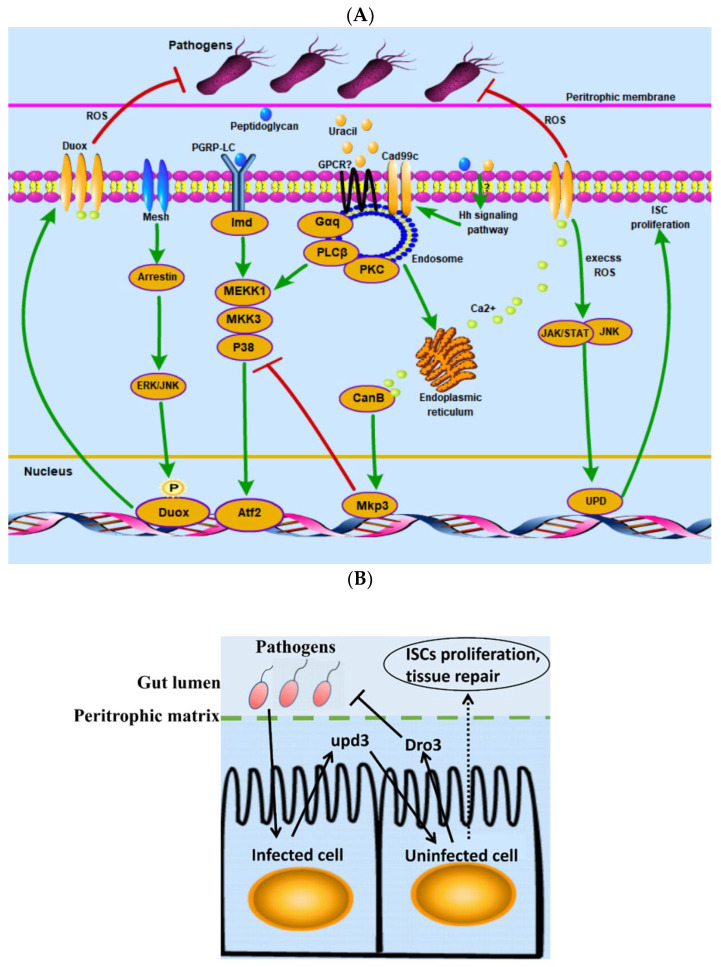
Regulation of ROS and the physiological roles of JAK/STAT signaling pathway in insect gut immunity. (**A**) Regulation of ROS (modified from the reference [59]): When pathogens invade the insect gut, secreted uracil stimulates the formation of Cad99C/PLCβ/PKC upstream signaling endosomes via GPCRs, which releases signaling molecules to regulate the flow of Ca^2+^ in the endoplasmic reticulum to activate the enzymatic activity of Duox in the cell membrane; Peptidoglycan also regulates the formation of Cad99C through the Hedgehog (Hh) signaling pathway to interact with in Duox enzyme. The transcriptional regulation of the Duox gene is regulated by the intranuclear transcription factor Atf2Atf2, which is regulated by the MEKK1/MEK3/p38 MAPK cascade; at the same time, the signal endosome activated by uracil activates the transcription of ATF2 through PLCβ, and the IMD signaling pathway also activates this pathway; MAKP phosphatase 3 (Mkp3) downregulates Duox expression, and PLCβ and calcium regulate neurophosphatase B (CanB) induce Mkp3 transcription;; Excess ROS in the intestine activates the JAK/STAT and JNK pathways, which transmit signals to the nucleus, enabling the expression of cellular repair signaling factor UPD that initiate repair of the intestine and thus participate in the intestinal immune defense response. (**B**) The physiological roles of JAK/STAT signaling pathway in insect gut immunity: When pathogens infect the intestine, upd3 ligands are produced in infected cells, which act as signaling molecules into uninfected healthy cells, activating Dro3 expression and intestinal repair. Upd3, the members of the Unpaired family; Dro3: Drosomycin 3.

**Table 1 ijms-23-15132-t001:** Physical barriers and key gut immunity response in different insects.

Species	Physical Barriers	Immunity Response
**Diptera:**		
*Drosophila melanogaster*	PM [1] Mucus [2]	Imd [3] Toll [4] Duox-ROS [5] JAK/STAT [6] NOX-ROS [7] Melanization [8]
*Bactrocera dorsalis*	PM [9]	Imd [10,11] Duox-ROS [12] JAK/STAT [9]
*Anopheles gambiae*	PM [13,14]	Imd [15] Duox-ROS [16] JAK/STAT [17]
**Lepidoptera:**		
*Bombyx mori*	PM [18]	Imd [19] Duox-ROS [20] JAK/STAT [20]
*Plutella xylostella*	PM [21]	Imd [22] Toll [22] JAK/STAT [22]
**Hemiptera:**		
*Acyrthosiphon pisum*	PMM [23]	JAK/STAT [24]
*Nilaparvata lugens*	PMM [25]	Imd [26] JAK/STAT [26]
**Coleoptera:**		
*Holotrichia oblita*	PM [27]	_
*Rhynchophorus ferrugineus*	_ _	Imd [28] Toll [29] JAK/STAT [30]
**Hymenoptera:**		
*Apis mellifera*	PM [31]	Imd [32] Duox-ROS [33] JAK/STAT [34] Melanization [35]
*Bombus muscorum*	_ _	Melanization [36]

“_” indicates that no studies on intestinal immunity have been reported in this species; “_ _” indicates that such species has not been reported to have PM or mucus.

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
