# Peer review of "The Intestinal Immune Defense System in Insects"

_ijms, 2022, doi:10.3390/ijms232315132_

Round 1
Reviewer 1 Report
--
Reviewer 2 Report
I’m sorry to say that I suggest to reject this review paper. I even can’t finish reading, since I found plagiarism in this paper. This is unethical. I suggest the editorial office detect this paper by plagiarism checking software.
This paper review insect defense from physical barriers, including peritrophic matrix and mucus layer, to immunity responses containing Imd, Duox-ROS, JAK/STAT signaling pathway, the structure is as same as Bai et al. (2020,https://doi.org/10.1111/1744-7917.12868), you can also find the same content in Bai et al.’s paper. Please check Figure 2 and the figure legend, you may find the figure is similar to Fig. 1 published by Bai et al.. The figure legend expressed the same meaning according to Bai et al. with many duplicate words. It seems that the authors would like to rewrite the existing review paper and resubmit it. In my opinion, such a paper would have little or no novelty to publish in this version.
Some suggestions can be found in the attached PDF file.

Reviewer 3 Report
Overall, the review article provides every detail about structure and function of insect’s gut and how immune response is activated and channeled in insects. However, there are few places where the article need correction.
1. I would like to suggest reframing some long sentences to short and avoid using repetitive words and proper punctuations could be used to explain it well. This can be done throughout the manuscript.
2. Spelling mistakes should be corrected throughout the paper, for example use one spelling for “defense”
3. In introduction, sentences are very long and no explanation about the role of PMM, and then started about PM, please discuss PM and PMM, separately in two sentences
4. Flow of the paper is missing in the article, lines in the paragraphs are not very well connected.
While reading the article it feels like paragraphs are written by different individuals.
5. There is problem in figure 2, text is hidden by some white box, it needs to be corrected in the manuscript.
6. Conclusion of the paper especially last paragraph should be framed nicely to explain the importance of insect’s immune response and some unknown questions which need to be addressed.
Round 2
Reviewer 2 Report
The paper reviewed insect intestinal immune defense system facing pathogenic microorganisms which is important for insect survival. I like the present title. I see that authors have revised the paper deeply. I only have one minor suggestion, please include IMD pathway in the abstract. The paper looks better now. I don’t need to review it again.
